# Stakeholders’ Perspective on the Key Features of Printed Educational Resources to Improve the Quality of Clinical Communication

**DOI:** 10.3390/healthcare12030398

**Published:** 2024-02-04

**Authors:** Silvia Gonella, Paola Di Giulio, Ludovica Brofferio, Federica Riva-Rovedda, Paolo Cotogni, Valerio Dimonte

**Affiliations:** 1Direction of Health Professions, City of Health and Science University Hospital of Torino, Bramante Avenue 88-90, 10126 Turin, Italy; 2Department of Public Health and Pediatrics, University of Torino, Santena Street 5 bis, 10126 Turin, Italy; paola.digiulio@unito.it (P.D.G.); ludovica.brofferio@edu.unito.it (L.B.); valerio.dimonte@unito.it (V.D.); 3Pain Management and Palliative Care, Department of Anesthesia, Intensive Care and Emergency, City of Health and Science University Hospital of Turin, University of Torino, Bramante Avenue 88-90, 10126 Turin, Italy; paolo.cotogni@unito.it

**Keywords:** clinical communication, co-creation, co-design, difficult conversation, education, manuals as topic, printed educational resources, professional practice, quality improvement, quality of healthcare

## Abstract

Social and healthcare professionals often feel ill equipped to effectively engage in difficult conversations with patients, and poor proficiency negatively affects the quality of patient care. Printed educational resources (PERs) that provide guidance on sustaining complex clinical communication may be a source of support if thoughtfully designed. This study aimed to describe the key features of PERs in order to improve the quality of clinical communication according to the perspective of meaningful stakeholders. This was a descriptive secondary analysis of data collected by three remote focus group discussions that involved 15 stakeholders in the context of developing an educational booklet to support professionals in complex communication scenarios. Focus groups were audio-recorded and transcribed verbatim, and an inductive thematic analysis was performed. Three key features of PERs that aim toward quality improvement in clinical communication were identified: (1) having the potential to provide benefits in clinical practice; (2) facilitating, encouraging, and enticing reading; and (3) meeting the need of professionals to improve or update their knowledge. These findings suggest that PERs relevant to professionals’ clinical priorities and learning needs may make their efforts to apply learning in practice more likely and consequently result in improved healthcare quality.

## 1. Introduction

Scientific and technological advances in medicine, such as new therapies and earlier screening tests, increasingly expose social and healthcare professionals (hereafter professionals) to complex communication scenarios regardless of their scope of practice and context of care [1].

Communication drives the process of care and the provision of quality care [2]. Effective communication is a key element to improve the quality of care throughout the disease path by alleviating anxiety, de-escalating conflicts, providing support, and improving professionals’ knowledge of patients’ care preferences to deliver care aligned with their preferences [3,4]. High-quality communication helps to collaboratively set an agenda, empathically respond to emotions, establish a mutual partnership, explore patients and families’ understanding about the clinical situation, and assess patients’ goals and priorities [5]. Instead, suboptimal communication negatively impacts patients, family carers, professionals, and the healthcare system, and is associated with a considerable portion of major adverse events [6]. About 60% of patients hospitalized in internal medicine wards make complaints in the domain of relationships, and when a clinical provider is named, complaints concerning communication and humaneness predominate [7].

Several barriers at multiple levels including patient/family carer-related factors, professional-related factors, and system-related factors can hinder the quality of communication [8]. Among the professional-related obstacles to effective communication, difficulties in prognostication, lack of time, comfort, and training are common [8,9], even if confidence can vary across specific professions and communication domains [10]. Professionals often experience a sense of being unprepared and lacking proficiency when they have to respond to strong patients’ or family carers’ emotions such as anger, fear, or grief, answer to unexpected comments, or deliver serious news; this also stems from the fact that communication skills training is often not part of their standard education [11,12,13]. Issues concerning confidence and training are relevant to their impact on patient care, as shown in a survey that involved over 4000 American physicians; those who felt more confident in discussing care choices with severely ill patients were more inclined to engage in conversations about prognosis, do-not-orders, hospice referral, and preferred place of death [14].

Many institutions have acknowledged the impact of effective communication on the delivery of high-quality care and invested in communication skills training programs. Most of these programs are accredited for continuing professional development, based on both didactic and experiential learning approaches, and employ several methods—printed educational resources (PERs), videos, small group discussions, e-learning self-education activities—to improve the professionals’ engagement rate, sustain their learning, and promote knowledge translation in daily practice [15,16]. Their ultimate goal is to enhance the quality and safety of patient care by promoting the development of knowledge and skills, and changes in attitudes and behaviors [17].

PERs have been largely employed alone or as part of multifaceted interventions to improve the quality of conversations that professionals offer to patients and their family carers [18]. PERs are relatively inexpensive low-tech solutions that are easy to implement and scale, and a recent Cochrane review found that they may improve the quality of professionals’ practice and patients’ outcomes; moreover, it seems that their computerized versions may make little or no difference compared to the same printed version [19]. In addition to learner-related factors such as learning style, educational needs, and motivation, attitudes, and beliefs, and professional standards for the desired behavior, other potential factors influencing the impact of PERs include their scientific soundness, format, layout, and design [19].

Educational resources that are designed without considering the perspective of stakeholders are rarely useful nor employed over time [20]. Therefore, exploring the lived experience of meaningful stakeholders in complex communication scenarios and what guidance they would need may help identify the key features to consider when designing PERs.

To improve the quality of communication in clinical encounters, we developed a communication quality improvement intervention, the Teach-to-Communicate Program. This is a multifaceted program that consists of (a) an educational booklet co-designed with meaningful stakeholders to support professionals in person-centered conversations with patients and their families; (b) experiential workshops based on improvisational theater techniques; (c) on-site training based on the educational booklet; (d) a community of practice where professionals have the opportunity to share problems that emerged during the clinical encounters and the strategies they employed; and (e) e-learning self-education activities.

In this manuscript, we report a qualitative analysis of the perspective of stakeholders that co-designed the educational booklet on the key features that the PERs aimed at improving the quality of clinical communication should address.

## 2. Materials and Methods

### 2.1. Study Design

This was a descriptive secondary analysis of data collected from focus group (FG) discussions in the context of developing an educational booklet to support professionals in complex communication scenarios [Manuscript under review]. The COnsolidated criteria for REporting Qualitative studies (COREQ) guidelines were followed to ensure methodological rigor [21]. The primary study was approved by the Ethics Committee of the University of Torino (n. 0598416/2021).

### 2.2. Primary Study

The primary study aimed to develop, validate, and preliminarily evaluate an educational booklet that provides guidance to professionals in difficult conversations.

The phase of booklet development entailed three FG discussions that were conducted in October 2022 and each session involved five participants. In total, 15 stakeholders (12 professionals with different scopes of practice in several care settings and 3 representatives of associations of patients, family carers, and volunteers in the field of dementia or palliative care) participated. Most participants were female and the mean working experience was 25 years (range 3–40) (Table 1).

The research team developed a first draft of the booklet. Then, the stakeholders were e-mailed the resource seven to ten days prior to the FG discussion and every participant was requested to read it so they could contribute ideas addressing its content, layout, or graphics during the FG discussion. All FGs took place remotely, with a member of the research team serving as moderator, and were guided by a predetermined set of questions that explored the booklet’s completeness, relevance to clinical practice, clarity, feasibility of use, and graphics. Moreover, a team member took field notes on participants’ body language. All participants provided their consent for the session to be recorded and actively engaged into the discussion by providing valuable comments that helped to guide the changes in the booklet. The comments that emerged during the discussions identified eight areas for improvement: (1) better frame the booklet as part of a communication skills training program; (2) improve the relational aspect of communication; (3) improve the realism of clinical scenarios; (4) emphasize the legal aspects; (5) add missing critical situations; (6) reword the booklet according to a person-centered dialogical approach; (7) improve actionability; and (8) improve layout and design. The booklet was adjusted according to the stakeholders’ feedback.

### 2.3. Secondary Analysis

The purpose of the secondary analysis was to identify the key features of PERs that aim to improve the quality of clinical communication based on the experience of meaningful stakeholders.

FG discussions were transcribed verbatim and S.G. checked transcripts for accuracy. Then, two researchers (L.B. and S.G.) independently analyzed the transcripts by employing an inductive thematic analysis approach [22]. This means that open coding was used with categories and themes emerging from the raw data through an interactive process [23]. L.B. and S.G met after coding the first FG transcript to discuss the developing coding sheets and resolve discrepancies. The final coding sheet was discussed within the research team and used for the analyses of the next two FG transcripts. L.B. and S.G worked together to group similar codes into higher level categories, which were restructured into overarching themes. The researchers followed a reiterative process of discussing agreements and disagreements to improve analytical rigor and achieve consensus. The research team was consulted when consensus was not reached. Categories, themes, and illustrative quotations were discussed and agreed within the research team. The qualitative analysis software ATLAS.ti version 8.4 aided the analysis.

Themes are exemplified through participants’ quotations, which are identified by a code indicating the FG and the stakeholders’ profile (e.g., FG2, internal medicine physician; FG3, social worker).

### 2.4. Trustworthiness

Guidelines for trustworthiness were followed [24]. To ensure credibility and dependability, the two researchers met after coding the first FG to consolidate codes, thereby improving reflexivity. Furthermore, all team members reviewed the final coding framework and consensus was reached on categories, themes, and illustrative quotations. Confirmability was ensured by keeping an audit trail of the entire data analysis and triangulation within the team to pinpoint categories, themes, and significant excerpts. Transferability was enhanced by detailing the data collection procedure and sample characteristics. Finally, authenticity was ensured by a well-planned online FG protocol that included choosing and piloting the interface, sending advance information about logistics, two observers monitoring the process, and the moderator inviting responses in the order that participants appeared on screen to minimize participants talking over one another [25]. Moreover, the FG moderator had extensive theoretical and clinical experience in difficult conversations and promoted a structured facilitation of group dialogue. This favored a secure, conducive online environment for in-depth discussion and the establishment of partnerships among participants [26].

## 3. Results

Overall, three themes that captured the key features of PERs aimed at improving the quality of clinical communication were identified: (1) having the potential to provide benefits in clinical practice; (2) facilitating, encouraging, and enticing reading; and (3) meeting the need of professionals to improve or update their knowledge (Table 2).

### 3.1. Theme 1: Having the Potential to Provide Benefits in Clinical Practice

All experts unanimously stated that PERs aimed at improving the quality of communication in clinical encounters should provide evidence-based and actionable guidance for patient-centered care. Such resources should support professionals in understanding patients’ needs and exploring patients’ preferences to align treatment accordingly, and offer a flexible reference model to address the main challenges of daily practice that professionals can adapt to the specific situation. This makes the resource potentially useful in different care settings and circumstances. Also, references to scientific literature (e.g., protocols and guidelines) help to highlight the essential elements that need to be implemented in clinical practice and promote reflection on how to deal with complex situations.

“*We cannot give examples of all the possible situations we daily face. The important thing is to have an outline to follow. [The resource is] a kind of grid that you can adjust to the situation you have to deal with*.”(FG2, internal medicine physician)

“*Each professional should have the opportunity to modulate the resource in the context where he/she works. It should be possible to extrapolate what it’s needed and make adaptations according to the organizational setting*.”(FG3, social worker)

To make the guidance more actionable, the resource should be quick to read and easy to use, provide space for taking notes, boxes summarizing the main content, and ready-to-use worksheets, and be available in different formats (e.g., pdf file or audio-recorded).

“*I’d love to have the opportunity to tear the page I need in that moment out. I could keep the page in my organizer or in my coat pocket and have it ready for use (…). This could make the written resource really useful in our everyday practice*.”(FG3, social worker)

When the care team accepts the resource as a working tool, the chances of long-term use increase and regular updates and revisions over time are required. Acceptance grows when resources reflect professionals’ priorities, are shared among the team members, and are part of structured educational interventions.

“*Introduction of new resources fails when they do not address professionals’ priorities or are not shared within the group. The use of resources ends when the group changes and they haven’t been recognized as useful for daily practice*.”(FG1, palliative care physician)

“*At some point, you may realize that [the resource] doesn’t work anymore (…) you need to review something. Otherwise, it would be like considering [the resource] as unchanging*.”(FG2, nursing home nurse)

“*I doubt that a resource can achieve the expected goals if professionals are not trained. Instead, it can represent the last phase of an educational intervention*.”(FG3, emergency physician)

According to the panel, the resource should be consistent with the setting of use and pay attention to the epidemiological, cultural, and organizational context. Moreover, the resource should be in agreement with the scope of practice of each professional of the care team and reflect the underpinning theoretical framework and philosophy of care:

“*Some sentences suggest a polarization between ‘doing’ and ‘not doing’, between ‘you can do more’ and ‘you can do less’ (…). According to me, it would be better to emphasise the importance of improving the quality of life and avoid the antithesis between ‘medicine that does’ and ‘medicine that does not*’.”(FG3, nurse representative of a palliative care association)

“*It’s not a technical issue, the issue is compassion, relationship, sharing of suffering. The issue is to sensitise people that is the pre-requirement for a relationship that is human and professional. Isn’t this the ultimate meaning that we must give to our work*?”(FG3, bioethicist)

The resource should also fit the normative context and adhere to ethical principles that guide clinical practice.

“*I would refer to the basis of ethics -autonomy, non-maleficence, beneficence and especially autonomy and justice- (…). According to the law 219/2017 [Provisions for informed consent and advance directives], when patients are not cognitively competent to share their care preferences, their families give voice to the preferences of their beloved. Families should be involved in the decision-making process but are not responsible for choosing. This is up to the healthcare professionals*.”(FG1, forensic physician)

### 3.2. Theme 2: Facilitating, Encouraging, and Enticing Reading

The panel discussed the role of graphic design, understandability of the content, and formal features such as typographical errors in promoting reading.

The experts highlighted the importance of graphics in shaping a clear structure that makes the resource easy to read and understand. Despite acknowledging the role of an appealing and consistent graphic throughout the resource, the panel also agreed that content has priority. Moreover, images and motifs should be relevant to the content and evoke its deeper meaning.

“*In my opinion, visuals with flowers in the background are confusing. The graphics should be functional to the meaning, if not… I can’t fully understand it. You can propose some warm images or reduce abstract forms or stylizations. However, this is not essential, because what needs to emerge is the content*.”(FG3, nurse representative of a palliative care association)

The experts engaged in an extensive discussion on how to balance the need to reflect the complexity of real-life practice and simplify the content to improve understandability and use. The panel agreed that being able to capture this dynamism can foster the acceptance and use of the resource. Simplification cannot be completely eliminated; however, it should be minimized. Additionally, too much detail as well as repetitions, clichés, and rhetoric need to be avoided.

“*Reality unfolds over a thousand nuances. It is not possible to define this in a booklet, however, if you could express this dynamic, the resource could probably be a little more heartfelt*.”(FG1, palliative care physician)

“*Content should not convey the message that everything has an answer in pills*.”(FG2, home palliative care nurse)

“*The guide should quickly lead me to what it’s important for my everyday practice*.”(FG3, emergency physician)

The panel identified several textual features that can entice reading, such as the absence of typographical errors, readable font and size, fluency, and simplicity of the language. The experts also emphasized the role of vocabulary, which should fit the aim of the resource. For example, the statement “manage emotions” was perceived as not appropriate for this resource because it conveys a professional-centered approach rather than a patient-centered approach.

“*The verb ‘manage’ should be replaced with another term, sentences need to provide guidance without being too directive. ‘Manage’ may be changed into ‘stand in front of’, so that the professional feels the responsibility to be in situation (…). When professionals manage the situation, it isn’t the ideal condition. Instead, it is when professionals embrace, listen, and act as a companion*.”(FG3, nurse representative of a palliative care Association)

### 3.3. Theme 3: Meeting the Need of Professionals to Improve or Update Their Knowledge

According to the experts, the resource should foster the development of professionals’ reflective skills by stimulating awareness of challenges in daily clinical practice and thoughts on the role and responsibilities of each team member. Professionals should reflect on their emotions before starting a clinical encounter to adopt an aware and attentive approach.

“*The resource should represent a way to improve attention, a more careful way of working*.”(FG2, home palliative care nurse)

The panel agreed that the resource should be consistent with adult education methods, give value to professionals’ previous experience, and reinforce and support professionals’ skills.

“*Adult education arises from the desire to do better for satisfying personal needs. Students in training don’t have this motivation. It would be good to take this dynamic of adult education into account*.”(FG1, palliative care physician)

“*[The resource] prompts us to restart something we have lost, especially when working in rushed hospital wards. If we don’t practice to work in a certain way, we won’t have the skills when time is enough.*”(FG2, internal medicine nurse)

## 4. Discussion

Professionals often lack knowledge of techniques and strategies to successfully engage in effective communication with patients and their family carers [12] and struggle to find guidance on how to communicate [10]. PERs remain one of the most common sources of information that professionals look at [19]. As suggested by several theories on quality improvement in patient care, there are several factors—cognitive, attitudinal, motivational, professional, and social—that may influence the impact of PERs [27]. Therefore, the involvement of meaningful stakeholders in the development of PERs is pivotal to shape educational material that is consistent with professional priorities and actionable in daily practice. This study identified three main key features of PERs that are aimed at improving the quality of clinical communication, according to the perspective of a diverse group of stakeholders: (1) having the potential to provide benefits in clinical practice; (2) facilitating, encouraging, and enticing reading; and (3) meeting the need of professionals to improve or update their knowledge.

The panel unanimously agreed that PERs aimed at improving the quality of clinical communication should be accepted working tools in the care team and fit the professionals’ scope of practice and the epidemiological, cultural, normative, and organizational context. This helps to keep the care team on the same page and facilitates care coordination [28]. Despite acceptability becoming a key element in the social and healthcare landscape, its definition is not straightforward. The theoretical framework of acceptability consists of seven constructs—affective attitude, burden, perceived effectiveness, ethicality, coherence, opportunity costs, and self-efficacy [29]—which all emerged in our FG discussions. Our experts highlighted that PERs should align with adult education principles and acknowledge the value of users’ prior experience (affective attitude), be easy to understand and use (burden), and have a good fit with the scope of practice of targeted users (ethicality) and their priorities (coherence). Moreover, the potential users should be clear about the benefits of the resource and the extent to which it is likely to achieve the intended purpose (opportunity costs and perceived effectiveness); finally, the resource should reinforce the users’ skills (self-efficacy).

To be useful in clinical practice, the panel agreed that PERs should provide a flexible, evidence-based model that professionals can adapt to the specific situation. Consistent with the previous literature [19], the experts highlighted the importance of tools that are anchored in scientific evidence but do not mandate sequences of actions. Although communication protocols are undoubtedly useful and can offer support, particularly to novices or professionals with limited experience, communication is not linear and cannot be squeezed into strict protocols. PERs that seek to improve the quality of communication should highlight the role of humaneness and interactivity in clinical encounters and provide guidance without being overly restrictive to allow professionals to be mindful during the clinical encounter [28]. Mindfulness requires professionals to be in-the-moment with the patient/family carer(s) and adapt to dynamic changes in the interaction; instead, following predetermined scripts leads to non-patient-centered, mindless interactions [30]. Beyond sustaining the professional’s ability to monitor the dynamic of each interaction, a flexible reference model can be applied across different care settings and scopes of practice; this is functional for the ubiquity of complex communication scenarios.

The ease of reading was another element that the panel perceived as key for PERs to be actionable in daily practice. Readability is influenced by at least four factors—content, style, structure, and design [31,32]—and all of these features emerged in our findings. The panel pinpointed the need to find the right balance between the complexity of real-life clinical scenarios and simplified content to shape a realistic but easy-to-use resource. In addition, the experts mentioned the importance of a clear structure and stylistic elements such as fluency, wording, font, and size. Finally, the panel highlighted the role of a conducive design that can help users to understand and use the information when it is consistent with the aim of the resource and avoids clichés and rhetoric [33]. A wise combination of visuals and text leads to a synergistic effect on learning [32]. Visuals are a compelling means of communication because human memory can remember up to 6.5 times more information when it is presented visually compared to text alone [32]. In general, higher readability results in shorter reading times, improved content retention, and increased text comprehension [34].

Our experts recommended that PERs aimed at improving the quality of clinical communication should promote reflective practice by fostering awareness about clinical challenges and the role and responsibilities of each member of the care team. Such resources should support professional development beyond sustaining changes in behavior since communication is both a skill and a way of being in relation to the other [35]. Reflection is an essential element of clinical practice since it aids professionals in creating meaning from the experience and identifying their learning needs [36]. Therefore, PERs should direct professionals in a continuous cycle of reflection to favor a more attentive and caring way of working, which is essential for improving the quality of care. Reflection before and during action allows professionals to take a listening position physically, psychologically, and emotionally, and to be present in the interaction; reflection on one’s actions allows one to identify what went well, what needs to be improved, and lessons learnt for the future [37].

### Limitations

Online FGs partially prevented the recording of field notes on body language compared to in-person discussions. However, these data would have not been informative for the aim of this study and online discussions facilitated participation from different geographical areas [38]. Additionally, we can reasonably postulate that in-person FGs would have not resulted in richer information as synchronous online and in-person modalities are comparable both in terms of quantity and diversity of data [39].

The mean duration of our FGs was slightly shorter compared to the optimal recommended duration (78 min vs 90 min) [40]. This may indicate that the discussions were not as in-depth as they could be, probably due to involving more participants in each session than the literature suggests (5 vs 3–4) [41]. However, the participation of more stakeholders was justified by the need to encompass professionals with different scopes of practice who work within a broad spectrum of clinical settings to prompt discussion.

## 5. Conclusions

PERs aimed at improving the quality of clinical communication are best placed to promote changes in professionals’ behavior when they encourage reading and meet the users’ need to improve or update their knowledge, and potential users perceive their benefits in clinical practice.

The acceptance of PERs by the care team is essential for their employment over time and is promoted when PERs reflect users’ priorities, provide evidence-based and actionable guidance, and promote reflective practice in a continuous cycle of learning. Also, graphical and stylistic issues should not be overlooked as they may have a conducive role in promoting the understandability and use of the information.

Our findings provide relevant and actionable guidance on how to optimize PERs aimed at improving the quality of clinical communication. They suggest that PERs should promote schema acquisition rather than provide restrictive guidance; the provision of flexible reference models reduces the memory workload and has the potential to enhance clinical communication in diverse healthcare settings. Moreover, researchers should consider early collaboration with art or graphic designers to communicate key messages in a clear, concise, and effective manner. Additionally, this study suggests that PERs should be introduced as part of structured educational interventions and that local leaders favor sharing among the care team to improve their acceptability, thus providing useful tips for successful implementation.

Future primary studies should provide a detailed description of the PERs, the process of development, and the evaluation of their readability and design, in addition to publishing the resource with the report. This would promote the ease of replication, comparison across studies, and analysis of potential effect modifiers.

## Figures and Tables

**Table 1 healthcare-12-00398-t001:** Characteristics of the participants.

Participants in the Focus Groups (*n* = 15)	N
**Female gender**	9
**Age**, years, mean (range)	50 (25–72)
**Overall working experience**, years, mean (range)	24.6 (3–40)
**Experience in the current service**, year, mean (range)	14.5 (3–30)
**Profile**	
Nurse	6
Physician	4
Social worker	2
Psychologist	1
Architect	1
Bioethicist	1
**Setting of care**	
Medicine	3
Palliative care	3
Nursing home	3
Association of patients, family carers, and volunteers	3
Supportive services *	2
Emergency	1

* Continuity healthcare service (*n* = 1); forensic medicine (*n* = 1).

**Table 2 healthcare-12-00398-t002:** Codes, categories, and themes.

Themes	Categories	Codes
**Having the potential to provide benefits in** **clinical practice**	Providing guidance for patient-centered care	Clear intent Providing guidance on exploring the patient’s care preferences and aligning treatment accordingly Providing guidance to deal with complex real-life situations Providing guidance on understanding patients’ needs Patient-centered not professional-centered care Showing a reference model that needs to be tailored to complex real-life situations Covering the main, challenging real-life situations Generalizing content to different care settings and situations Evidence-based and clear content
	Being actionable	Quick to read Include ready-to-use worksheets Availability of space for user notes Usability Being available in different formats Presence of a summary box
	Accepted as a working tool	Regularly updated over time Reflecting the professionals’ priorities Being shared among team members Associated with structured educational interventions
	Consistent with the context of use	Consistent with the scope of practice of each professional Consistent with the cultural context Consistent with the normative context Consistent with the healthcare setting Consistent with the ethical standards Consistent with the epidemiological landscape Consistent with the theoretical framework of reference Consistent with the philosophy of care
**Facilitating, encouraging, and enticing reading**	Attractive and conducive graphic design	Clear structure Appealing graphics Consistency throughout the text Graphically well organized Images and motifs consistent with and relevant to the content Attention to graphical details Prioritizing content over graphics
	Balanced content between complexity and oversimplification	Conveying a sense of complexity Realism of cases and dialogues No redundancies in content No rhetoric or clichés Not too much detail No unnecessary simplifications
	Attention to textual features	Attention to wording Readability Simplicity Sobriety Attention to typographical errors Fluent text
**Meeting the need of professionals to improve or update their skills or knowledge**	Fostering professional’s reflective skills	Promoting reflection on the role and responsibilities of each team member Fostering professionals’ awareness of challenges in daily care and strategies for overcoming them
	Recognizing professional experience as a pillar	Consistent with the andragogy learning theory Reflecting the professionals’ experience Reinforcing and supporting the skills that professionals already possess

## Data Availability

Research data are available from S.G.

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
