# Peer review of "Stakeholders’ Perspective on the Key Features of Printed Educational Resources to Improve the Quality of Clinical Communication"

_healthcare, 2024, doi:10.3390/healthcare12030398_

Round 1
Reviewer 1 Report
Comments and Suggestions for Authors
REVIEW REPORT ON MANUSCRIPT: "STAKEHOLDERS’ PERSPECTIVE ON THE KEY FEATURES OF PRINTED EDUCATIONAL RESOURCES TO IMPROVE THE QUALITY OF CLINICAL COMMUNICATION".
Thank you for the opportunity to review this manuscript.
Brief summary
The primary aim of the manuscript was to describe the key features of printed educational resources, which aimed to improve the quality of clinical communication according to the perspective of stakeholders that have the potential to enhance the quality of clinical communication. The manuscript successfully identifies and discusses key features deemed crucial by stakeholders in enhancing the quality of clinical communication.
General comments
The authors demonstrate a thorough understanding of the existing literature on clinical communication and educational resources, providing a strong foundation for the study.
The manuscript makes several noteworthy contributions to the existing body of knowledge. Firstly, it provides a comprehensive overview of the key features that stakeholders consider crucial in printed educational resources. The identification of these features is vital for developing targeted and effective educational materials in clinical communication.
Secondly, the manuscript adds value by presenting a nuanced analysis of the diverse perspectives of stakeholders, including nurses, physicians, social workers, psychologists and architect bioethicists. Understanding the varied needs and expectations of these groups is paramount for tailoring educational resources to meet the requirements of different stakeholders involved in clinical communication.
Thirdly, the manuscript contributes to the ongoing discourse on the importance of printed materials in the era of digital communication. By focusing on printed educational resources, the research sheds light on the enduring relevance and impact of traditional forms of communication within the clinical setting.
SPECIFIC COMMENTS
1. In page 2, line 42-44, the keywords are too many. Authors should revise it.
2. In page 4, line 138, Table 1, authors should include other characteristics of the participants, such as gender should be described to enable readers understand the participants. I suggest the table should be revised.
3. In page 13, line 415, the authors should discuss the practical implications of the findings, emphasizing their relevance for improving the quality of clinical communication through tailored educational resources. They should also elaborate on the implications of their findings for clinical education, policy, and recommendation for further research studies.
In conclusion, the manuscript contributes significantly to the field of clinical communication by describing stakeholders' perspectives on printed educational resources. The comprehensive analysis and clear presentation of findings enhance its scholarly merit. However, minor revisions related to the clarity of certain sections and additional discussion on implications findings could further strengthen the manuscript. Overall, the research has the potential to inform the development of effective educational materials to enhance clinical communication in diverse healthcare settings.
Author Response
We would thank the reviewer for his(er) words of appreciation. Changes in the text have been highlighted in yellow to be easily trackable.

Reviewer 2 Report
Comments and Suggestions for Authors
Thank you for this interesting qualitative analysis of PERs. The topic is of interest in health communications. I have a few suggestions which are intended to increase the quality of the submission.
Methods:
The methods are unclear in terms of the stakeholders and their role in co-designing the PER booklet. Line 108 indicates that the stakeholders designed the booklet - does that pertain to all 15 stakeholders? Did all 15 stakeholders contribute equally in their co-design role? Please provide more specifics on what/how the stakeholders contributed to creation of the booklet.
Line 129: the stakeholders were provided the booklet and asked to "read the booklet....and contribute ideas to the content, layout, graphics, etc. when they were in their Focus Group discussions. This is confusing to the reader, as previously (line 108) we are told the stakeholders co-designed the booklet, and now in line 129 we are told they should contribute ideas to the content, layout, graphics, etc. Wouldn't that be co-designing the booklet? What booklet then are they reading (line 129) and then being asked to comment on? One that they previously created? Please clarify this section, as it is not clear to the reader.
Line 136: comments emerged to guide changes to the booklet. It is not clear if the booklet was changed for the Focus Group discussions, or the changes will be planned for the future. Please clarify.
In the results and discussion, there is mention as to the importance of graphics and visuals. This is an important aspect to be expounded upon. Perhaps you can glean some insights from this reference, and add to your discussion.
Barlow B, Webb A, Barlow A. Maximizing the visual translation of medical information: A narrative review of the role of infographics in clinical pharmacy practice, education, and research .J Am Coll Clin Pharm. 2021;4:257–266.https://doi.org/10.1002/jac5.1386266
Comments on the Quality of English Language
The authors' intent was at times difficult to follow due to English language concerns (grammar, context, etc).
Author Response
Thank you for your helpful comments that helps us to improve the quality of the manuscript. Changes in the text have been highlighted in yellow to be easily trackable.
